# FAST WEIGHT LONG SHORT-TERM MEMORY

**T. Anderson Keller, Sharath Nittur Sridhar, Xin Wang**
Intel AI Lab, Artificial Intelligence Products Group, Intel Corporation
{andy.a.keller, sharath.nittur.sridhar, xin3.wang}@intel.com

## ABSTRACT

Associative memory using fast weights is a short-term memory mechanism that substantially improves the memory capacity and time scale of recurrent neural networks (RNNs). As recent studies introduced fast weights only to regular RNNs, it is unknown whether fast weight memory is beneficial to gated RNNs. In this work, we report a significant synergy between long short-term memory (LSTM) networks and fast weight associative memories. We show that this combination, in learning associative retrieval tasks, results in much faster training and lower test error, a performance boost most prominent at high memory task difficulties.

## 1 INTRODUCTION

RNNs are highly effective in learning sequential data. Simple RNNs maintain memory through hidden states that evolve over time. Keeping memory in this simple, transient manner has, among others, two shortcomings. First, memory capacity scales linearly with the dimensionality of recurrent representations, limited for complex tasks. Second, it is difficult to support memory at diverse time scales, particularly challenging for tasks that requires information from variably distant past.

Numerous differentiable memory mechanisms have been proposed to overcome the limitations of deep RNNs. Some of these mechanisms, e.g. attention, have become a universal practice in real-world applications such as machine translation (Bahdanau et al., 2014; Daniluk et al., 2017; Vaswani et al., 2017). One type of memory augmentation of RNNs includes mechanisms that employ long-term, generic key-value storages (Graves et al., 2014; Weston et al., 2015; Kaiser et al., 2017). Another kind of memory mechanisms, inspired by early work on fast weights (Hinton & Plaut, 1987; Schmidhuber, 1992), uses auto-associative, recurrently adaptive weights for short-term memory storage (Ba et al., 2016a; Zhang & Zhou, 2017; Schlag & Schmidhuber, 2017). Associative memory considerably ameliorates limitations of RNNs. First, it liberates memory capacity from the linear scaling with respect to hidden state dimensions; in the case of auto-associative memory like fast weights, the scaling is quadratic (Ba et al., 2016a). Neural Turing Machine (NTM)-style generic storage can support memory access at arbitrary temporal displacements, whereas fast weight-style memory has its own recurrent dynamics, potentially learnable as well (Zhang & Zhou, 2017). Finally, if architected and parameterized carefully, some associative memory dynamics can also alleviate the vanishing/exploding gradient problem (Dangovski et al., 2017).

Besides memory augmentation, another entirely distinct approach to overcoming regular RNNs' drawbacks is by clever design of recurrent network architecture. The earliest but most effective and widely adopted one is gated RNN cells such as long short-term memory (LSTM) (Hochreiter & Schmidhuber, 1997). Recent work has proposed ever more complex topologies involving hierarchy and nesting, e.g. Chung et al. (2016); Zilly et al. (2016); Ruben et al. (2017).

How do gated RNNs such as LSTM interact with associative memory mechanisms like fast weights? Are they redundant, synergistic, or rather competitive to each other? This remains an open question since all fast weight networks reported so far are based on regular, instead of gated, RNNs. Here we answer this question by revealing a strong synergy between fast weight and LSTM.

## 2 RELATED WORK

Our present work builds upon results reported by Ba et al. (2016a), using the same fast weight mechanism. A number of studies subsequent to Ba et al. (2016a), though not applied to gated RNNs, pro-

posed interesting mechanisms directly extending or closely related to fast weights. WeiNet (Zhang & Zhou, 2017) parameterized the fast weight update rule and learned it jointly with the network. Gated fast weights (Schlag & Schmidhuber, 2017) used a separate network to produce fast weights for the main RNN and the entire network was trained end-to-end. Rotational unit of memory (Dangovski et al., 2017) is an associative memory mechanism related to yet distinct from fast weights. Its memory matrix is updated with a norm-preserving operation between the input and a target.

Danihelka et al. (2016) proposed an LSTM network augmented by an associative memory that leverages hyperdimensional vector arithmetic for key-value storage and retrieval. This is an NTM-style, non-recurrent memory mechanism and hence different from the fast weight short-term memory.

## 3 FAST WEIGHT LSTM

Our FW-LSTM is defined by the following update equations for the cell states, hidden state, and fast weight matrix (Figure 1).[1]

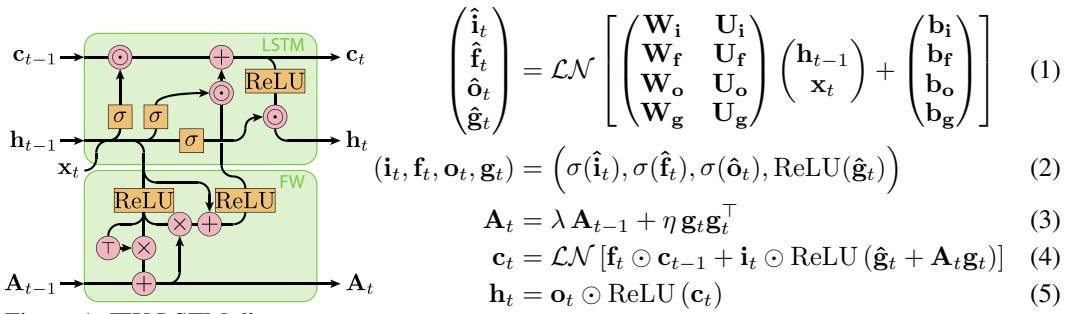

$$\begin{pmatrix}\hat{\mathbf{i}}_t \\ \hat{\mathbf{f}}_t \\ \hat{\mathbf{o}}_t \\ \hat{\mathbf{g}}_t\end{pmatrix} = \mathcal{LN}\left[\begin{pmatrix}\mathbf{W_i} & \mathbf{U_i} \\ \mathbf{W_f} & \mathbf{U_f} \\ \mathbf{W_o} & \mathbf{U_o} \\ \mathbf{W_g} & \mathbf{U_g}\end{pmatrix}\begin{pmatrix}\mathbf{h}_{t-1} \\ \mathbf{x}_t\end{pmatrix} + \begin{pmatrix}\mathbf{b_i} \\ \mathbf{b_f} \\ \mathbf{b_o} \\ \mathbf{b_g}\end{pmatrix}\right] \quad (1)$$

$$(\mathbf{i}_t, \mathbf{f}_t, \mathbf{o}_t, \mathbf{g}_t) = \left(\sigma(\hat{\mathbf{i}}_t), \sigma(\hat{\mathbf{f}}_t), \sigma(\hat{\mathbf{o}}_t), \mathrm{ReLU}(\hat{\mathbf{g}}_t)\right) \quad (2)$$

$$\mathbf{A}_t = \lambda\,\mathbf{A}_{t-1} + \eta\,\mathbf{g}_t\mathbf{g}_t^\top \quad (3)$$

$$\mathbf{c}_t = \mathcal{LN}\left[\mathbf{f}_t \odot \mathbf{c}_{t-1} + \mathbf{i}_t \odot \mathrm{ReLU}\left(\hat{\mathbf{g}}_t + \mathbf{A}_t\mathbf{g}_t\right)\right] \quad (4)$$

$$\mathbf{h}_t = \mathbf{o}_t \odot \mathrm{ReLU}\left(\mathbf{c}_t\right) \quad (5)$$

Figure 1: FW-LSTM diagram

Here $\mathbf{x}_t \in \mathbb{R}^d$, $\mathbf{h}_t, \mathbf{v}_t, \hat{\mathbf{v}}_t, \mathbf{b_v} \in \mathbb{R}^h$, $\mathbf{W_v}, \mathbf{A}_t \in \mathbb{R}^{h\times h}$ and $\mathbf{U_v} \in \mathbb{R}^{h\times d}$, where $\mathbf{v} \in \{\mathbf{i}, \mathbf{f}, \mathbf{o}, \mathbf{g}\}$, and $t$ indexes time steps. $\odot$ denotes Hadamard (element-wise) product, $\mathcal{LN}[\cdot]$ layer normalization, and $\sigma(\cdot)$, $\mathrm{ReLU}(\cdot)$ are the sigmoind and rectified linear function applied element-wise. We used $\mathrm{ReLU}(\cdot)$ in places of $\tanh(\cdot)$ for efficiency, as it did not make a significant difference in practice. Our construction is identical to the standard LSTM cell except for a fast weight memory $\mathbf{A}_t$ queried by the input activation $\mathbf{g}_t$. Since $\mathbf{g}_t$ is a function of both the network output $\mathbf{h}_{t-1}$ and the new input $\mathbf{x}_t$, this gives the network control over what to associate with each new input.

## 4 EXPERIMENTS

To study the performance of FW-LSTM in comparison with FW-RNN and LSTM with layer normalization (LN-LSTM), we experimented with the associative retrieval task (ART) described in Ba et al. (2016a). Input sequences are composed of $\frac{K}{2}$ key-value pairs followed by a separator ??, and then a query key, e.g. for $K = 8$, an example sequence is a1b2c3d4??b whose target answer is 2. We experimented with sequence lengths much greater than the original $K = 8$, up to $K = 30$ similar to Zhang & Zhou (2017) and Dangovski et al. (2017).

We further devised a modified ART (mART) that is a re-arrangement of input sequences in the original ART. In mART, all keys are presented first, then followed by all values in the corresponding order, e.g. the mART equivalent of the above training example is abcd1234??b with target answer of again 2. In contrast to ART, where the temporal distance is constantly 1 between associated pairs and only average retrieval distance grows with $K$, in mART temporal distances of both association and retrieval scales linearly with $K$. This renders the task more difficult to learn than the original ART, and $K$ can be used to control the difficulty of memory associations.

In all experiments, we augmented the FW-LSTM cell with a learned 100-dimensional embedding for the input $\mathbf{x}_t$. Additionally, network output at the end of the sequence was processed by another

---

[1] Note that the placement of layer normalizations is slightly different from the method described in the original paper (Ba et al., 2016b) We find applying layer normalization to the hidden state and input activations simultaneously (rather than separately as in the original model) worked better for this fast weight architecture.

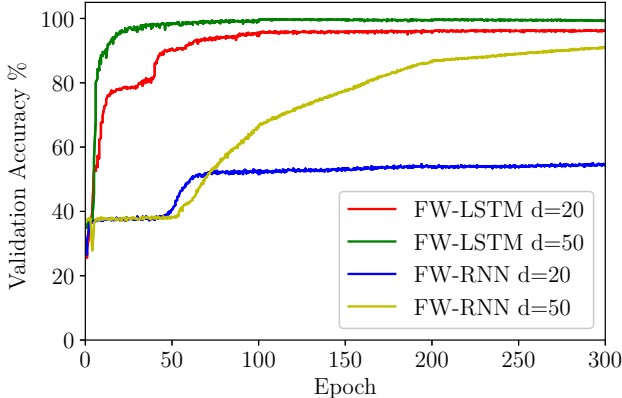

Figure 2: Validation accuracy during the course of training of mART for FW-LSTMs and FW-RNNs of 20 and 50 hidden units.

hidden layer with 100 ReLU units before the final softmax, identical to Ba et al. (2016a). All models were tuned as described in **Appendix** and run for a minimum of 300 epochs. [2]

The left half of Table 1 shows performances of LN-LSTM, FW-RNN[3], and our FW-LSTM trained on ART with different sequence lengths and numbers of hidden units. FW-LSTM has a slight advantage when the number of hidden units is low, but otherwise both the FW-RNN and FW-LSTM solve the task perfectly.

The right half of Table 1 shows performances of the same models trained on the mART. Due to significantly increased difficulty of the task, we instead show results for sequence lengths $K = 8, 16$. In learning mART, FW-LSTM outperformed FW-RNN and LN-LSTM by a much greater margin especially at high memory difficulty, $K = 16$, and also converged much faster (Figure 2).

Table 1: Test accuracy (%) of associative retrieval (ART) and modified associative retrieval (mART) for different sized models and sequence lengths $K$.

| Task | | ART | | mART | | |
|---|---|---|---|---|---|---|
| **# Hidden** | **Model** | K = 8 | K = 30 | K = 8 | K = 16 | **# Parameters** |
| | LN-LSTM | 37.8 | 22.7 | 38.2 | 29.5 | 19k |
| 20 | FW-RNN | 98.7 | 95.7 | 55.5 | 30.3 | 12k |
| | FW-LSTM | **99.6** | **97.5** | **96.3** | **38.9** | 19k |
| | LN-LSTM | 95.4 | 21.0 | 34.8 | 25.7 | 43k |
| 50 | FW-RNN | **100.0** | **100.0** | 90.9 | 29.0 | 20k |
| | FW-LSTM | **100.0** | **100.0** | **99.4** | **93.3** | 43k |
| | LN-LSTM | 97.6 | 18.4 | 33.4 | 22.5 | 100k |
| 100 | FW-RNN | **100.0** | **100.0** | 91.9 | 30.5 | 38k |
| | FW-LSTM | **100.0** | **100.0** | **99.9** | **92.6** | 100k |

## 5 CONCLUSIONS

We observed that FW-LSTM trained significantly faster and achieved lower test error in performing the original ART. Further, in learning the harder mART, when input sequences are longer, we found that FW-LSTM could still perform the task highly accurately, while both FW-RNN and LN-LSTM utterly failed. This was true even when FW-LSTM had fewer trainable parameters. These results suggest that gated RNNs equipped with fast weight memory is a promising combination for associative learning of sequences.

---

[2]All code to reproduce the experimental results is available at [github link available after acceptance].

[3] The parameters $\eta$ and $\lambda$ used for FW-RNN here are different than those in Zhang & Zhou (2017), resulting in an improved performance. The values used are listed in **Appendix**.

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

## 6 ACKNOWLEDGMENTS

We thank Drs. Tristan J. Webb, Marcel Nassar and Amir Khosrowshahi for insightful discussions. We also thank Dr. Jason Knight for his assistance setting up Kubernetes cluster used for training and tuning.

## 7 APPENDIX

### 7.1 ASSOCIATIVE RETRIEVAL HYPERPARAMETERS

All models in the Associative retrieval section were tuned over the following hyperparameter ranges using standard grid search. The final models were selected based on the highest validation set accuracy from the following set:

$$\eta \in \{1.0, 0.75, 0.5, 0.25, 0.1\} \tag{6}$$
$$\lambda \in \{0.99, 0.9\} \tag{7}$$
$$\texttt{grad\_clip} \in \{1.0, 5.0\} \tag{8}$$
$$\texttt{learning\_rate} \in \{10^{-4}, 10^{-5}\} \tag{9}$$
$$\texttt{anneal\_rate} \in \{100, 10\} \tag{10}$$
$$\tag{11}$$

where $\texttt{anneal\_rate}$ is the number of epochs between which the learning rate is halved, and $\texttt{grad\_clip}$ is the maximum L2 norm clipping value for the gradient.

The optimal hyperparameters were found to match for both the FW-RNN and FW-LSTM for the simple associative retrieval tasks. They are as follows:

$$\eta = 1.0 \tag{12}$$
$$\lambda = 0.99 \tag{13}$$
$$\texttt{grad\_clip} = 5.0 \tag{14}$$
$$\texttt{learning\_rate} = 10^{-4} \tag{15}$$
$$\texttt{anneal\_rate} = 100 \tag{16}$$
$$\tag{17}$$

