# OpenReview forum: "Fast Weight Long Short-Term Memory"
_ICLR.cc/2018/Workshop — Reject_

### Official Review · AnonReviewer3 · 2018-03-09
**Applying "Fast Weight to Attend to the Recent Past" to LSTMs**

**Rating:** 5
**Confidence:** 3

**Review:**

This submission explores whether the formulation of attention proposed in "Using Fast Weights to Attend to the Recent Past" paper is applicable to LSTM. The associative memory bit is applied to the proposed input (often denoted by "j") before it's gated ("i") and added to the gated state ("f*c_{t-1}").

The experiments are done on a simple associative retrieval task and a new, harder variant of the same. The results indicate that the fast weight LSTM improves on both the plain LSTM and the fast weight RNN. It is unclear how trustworthy these results are since the experimental setup (especially hyperparameter tuning) is not described in much detail.

The proposed model is simple, the experiments are a good start, but even for a workshop paper, I feel more convincing ones are necessary.

---

> ### Public Comment · ~Thomas_Anderson_Keller1 · 2018-03-22
> **Reviewer 3 Response**
>
> Dear Reviewer 3,
>
> Thank you for taking the time to review our work.
>
> We believe there is a misunderstanding related to our description of the experimental setup since we have exhaustively described the setup in the paper. We indeed did a systematic grid search to optimize hyperparameters, to which the entire Appendix is dedicated. Additionally, as mentioned in the paper, with the intended release of the code, our results will be transparent and reproducible, significantly mitigating any uncertainty.
>
> We believe our results, despite simplicity due to usage of toy examples, are categorical in that they demonstrated a task which two existing models utterly failed to learn, but can be almost perfectly solved when the two mechanisms were combined.  This has never been reported before.
>
> Regards

---

### Official Review · AnonReviewer2 · 2018-03-10
**Incorporating Fast Weights into the LSTMs**

**Rating:** 5
**Confidence:** 3

**Review:**

Summary: This paper proposes to incorporate fast weights to the LSTMs. In a way the proposed method resembles WeiNet's construction by Zhang&Zho which is in turn inspired by the Ba et al's work on fast weights. The authors test their model on associative retrieval tasks and they show better results than FW-RNN and LN-LSTM. Overall, the paper is well-written.

Criticism:
- The results are quite encouraging, however it would have been much more convincing if they have shown some results on non-toyish tasks. It is difficult to say if their results would hold on more realistic tasks.
- The comparisons against other memory models such as DNCs or LSTMS are missing.
- Concept is interesting, but the novelty of the model in this paper is not that big. This would not be an issue if they had results on a real world task.

---

> ### Public Comment · ~Thomas_Anderson_Keller1 · 2018-03-22
> **Reviewer 2 Response**
>
> Dear Reviewer 2,
>
> Thank you for your thoughtful review.
>
> We are working on a demonstration of solutions to real-world problems using FW-LSTM. However, we believe our results using toy examples in this short paper are still compelling and useful as a starting point for others to begin experimenting with this new architecture. Specifically we think the novelty of the proposed model is undervalued when combined with the modified associative retrieval task which demonstrates, and provides insights into, its advantages.
>
> Our results in fact included direct comparison against LSTM and FW-RNN, and showed that either mechanism alone failed to learn a task while the combination of the two succeeded. Our goal is to demonstrate this synergy conclusively and concisely in this short paper, and thus we did not compare and contrast against other memory mechanisms like DNCs. Additionally, the LSTM results were included in Table 1, but validation accuracy curves were omitted from Figure 2 due to their significantly lower final validation error. These can easily be added for completeness if desired.
>
> We believe our work is novel in the sense that this is by far the first attempt to combine gated RNN cells with memory augmentations of RNNs, which are two approaches to enhancing RNNs that, to this day, have only been used in separation. Most importantly, our results demonstrated a task that either of the approaches alone failed to learn, but could be successfully solved by their combination.
>
> Regards

---

### Official Review · AnonReviewer1 · 2018-03-10
**Study on using fast weights on LSTMs**

**Rating:** 6
**Confidence:** 3

**Review:**

The authors study the application of fast weights to LSTM cells. Fast weights have been shown to improve the memory capacity and time scale of RNNs. In this study the authors study whether fast weights can help gated RNNs like LSTM cells. The conclusion of this work is that fast weights is also beneficial to LSTMs and results in much faster training and lower test error.

---

> ### Public Comment · ~Thomas_Anderson_Keller1 · 2018-03-22
> **Reviewer 1 Response**
>
> Dear Reviewer 1,
>
> Thank you for your review. We agree with your understanding of the paper and appreciate the time spent to understand our work.

---

### Decision · Program_Chairs · 2018-03-20
**ICLR 2018 Workshop Acceptance Decision**

**Decision:**

Reject

**Comment:**

Based on the reviews, this paper has not been accepted for presentation at the ICLR workshop. However, the conversation and updates can continue to appear here on OpenReview.